# Neuropathological Features of Gaucher Disease and Gaucher Disease with Parkinsonism

**DOI:** 10.3390/ijms23105842

**Published:** 2022-05-23

**Authors:** Makaila L. Furderer, Ellen Hertz, Grisel J. Lopez, Ellen Sidransky

**Affiliations:** Medical Genetics Branch, National Human Genome Research Institute, National Institutes of Health, 35A Convent Drive-Room 1E623, Bethesda, MD 20892-3708, USA; makaila.furderer@nih.gov (M.L.F.); ellen.hertz@nih.gov (E.H.); glopez@mail.nih.gov (G.J.L.)

**Keywords:** neuropathology, Gaucher disease, Parkinson disease, Lewy Body disorder, lysosomal storage disorder, Lewy body, glucocerebrosidase

## Abstract

Deficient acid β-glucocerebrosidase activity due to biallelic mutations in *GBA1* results in Gaucher disease (GD). Patients with this lysosomal storage disorder exhibit a wide range of associated manifestations, spanning from virtually asymptomatic adults to infants with severe neurodegeneration. While type 1 GD (GD1) is considered non-neuronopathic, a small subset of patients develop parkinsonian features. Variants in *GBA1* are also an important risk factor for several common Lewy body disorders (LBDs). Neuropathological examinations of patients with GD, including those who developed LBDs, are rare. GD primarily affects macrophages, and perivascular infiltration of Gaucher macrophages is the most common neuropathologic finding. However, the frequency of these clusters and the affected anatomical region varies. GD affects astrocytes, and, in neuronopathic GD, neurons in cerebral cortical layers 3 and 5, layer 4b of the calcarine cortex, and hippocampal regions CA2–4. In addition, several reports describe selective degeneration of the cerebellar dentate nucleus in chronic neuronopathic GD. GD1 is characterized by astrogliosis without prominent neuronal loss. In GD-LBD, widespread Lewy body pathology is seen, often involving hippocampal regions CA2–4. Additional neuropathological examinations in GD are sorely needed to clarify disease-specific patterns and elucidate causative mechanisms relevant to GD, and potentially to more common neurodegenerative diseases.

## 1. Introduction

Gaucher disease (GD) is a lysosomal storage disorder resulting from mutations in the *GBA1* gene that lead to decreased activity of acid β-glucocerebrosidase (GCase, E.C 3.2.1.45.). This enzyme cleaves the lipids glucocerebroside (GluCer) into glucose and ceramide [1], and glucosylsphingosine (GluSph) into glucose and sphingosine. Failure of this enzyme to clear these substrates from lysosomes causes macrophages to become engorged with lipid, giving rise to what are known as “Gaucher cells”. Typically, GD has been subdivided into three types based on presence and rate of progression of neurological involvement. However, GD can also be seen as a phenotypic spectrum due to the diversity of associated clinical manifestations, with the primary distinction being the degree of central nervous system (CNS) involvement [2]. Type 1, or non-neuronopathic GD (GD1), has presentations ranging from asymptomatic adults to young patients with significant visceral or skeletal disease. The most severe type, acute neuronopathic or type 2 (GD2), is associated with progressive neurodegeneration and early lethality. The disease manifests before 6 months of age, and some cases may present perinatally with congenital ichthyosis or hydrops fetalis [3,4,5,6]. Type 3 (GD3), or chronic neuronopathic GD, has neurologic involvement, particularly the presence of oculomotor involvement, that typically presents in early childhood with a slower progression. Even within GD3 there are multiple different phenotypes. Some patients have remarkable visceral and skeletal involvement with few neurological manifestations, while others may have learning disabilities, autism, generalized seizures, or progressive myoclonic epilepsy (PME) [2]. In addition, slowing of the horizontal saccadic eye movements, discrepant verbal and performance IQ scores, and background slowing on EEG are frequently observed in GD3 [3,7,8,9].

Over two decades ago it was appreciated that a small subset of adult patients with GD also develop parkinsonian features [10,11]. Greater awareness of patients sharing these disorders led both to the identification of further patients in GD clinics around the world [10], and the observation that parkinsonism was also more frequent among relatives of GD probands [12]. Subsequently, patients diagnosed with sporadic Parkinson disease (sPD) were also found to carry pathologic heterozygous variants in *GBA1* [2,13]. Ultimately, large multicenter studies confirmed that heterozygous *GBA1* mutations is a genetic risk factor for both Parkinson disease (PD) and dementia with Lewy bodies (DLB), increasing the disease risk 5–10-fold, depending on the specific mutation [14,15]. The risk of developing parkinsonism for patients with GD is not well established and varies between studies, from 9–12% at the age of 80 [16,17] to a 20-fold increased lifetime risk [18]. Importantly, a majority of GD1 patients never exhibit parkinsonian features, indicating that there is a more complex interplay underlying the neurodegeneration. In cell and animal models of GD, GCase deficiency is accompanied by neuroinflammation, evident by glial activation, as well as α-synuclein (α-syn) accumulation [19,20]. However, the exact mechanism underlying *GBA1*-associated PD remains unknown. Hypotheses include both gain-of-function due to promotion of α-syn aggregation, and loss-of-function leading to neurotoxic lipid accumulation, as well as a bidirectional feedback loop between GCase activity and α-syn aggregation, although no theory has been fully validated [19,21].

To better understand the disease pathogenesis, we reviewed the neuropathological features associated with glucocerebrosidase deficiency, examining autopsy studies of rare patients with GD. The limited number of cases, especially in subjects with non-neuronopathic GD, highlight the need for standardization of examinations. In addition, we examined reports of neuropathologic studies conducted on patients with GD who developed parkinsonism and compared the findings to heterozygous *GBA1*-mutation carriers with parkinsonism, which are more frequently examined. As uncertainty persists regarding the mechanism underlying *GBA1*-associated synucleinopathy, an evaluation of neuropathological features associated with GCase deficiency, could provide clues into pathways contributing to the clinical features observed.

## 2. Specific Brain Regions Are Involved in Gaucher Disease

Published reports of neuropathological evaluations of patients with GD are few, and most have been sporadic case studies including autopsy findings. The first larger and more comprehensive evaluation of the neuropathology of GD was published in 2004, examining autopsies of 12 patients with all three types of GD [22]. Certain neuropathological features of GD were shared among individuals with each of the three types. The cell types most often affected were neurons and astrocytes, and distinct regional specificity was noted [22]. The cerebral cortical layers 3 and 5, layer 4b of the calcarine cortex, and hippocampal areas CA2–4 were selectively involved in all forms of GD, although the extent of the abnormalities seen appeared to be dependent on the severity of the disease [22]. Regions adjacent to the specific areas involved, including the hippocampal CA1 region and calcarine lamina 4a and 4c, were spared, emphasizing the specificity of neural involvement (Figure 1). The authors also demonstrated that in wildtype brain the pyramidal neurons in CA2–4 and cortical layer 5 showed intense anti-GCase immunoreactivity, suggesting that this region might be especially vulnerable to diminished GCase levels [22]. Generally, a low level of background gliosis was observed, which was associated with the vasculature and most apparent in the brainstem and striatum. Perivascular clusters of Gaucher cells were identified in all cases, with a generally higher disease burden in GD2 and GD3 compared to GD1.

## 3. Neuropathological Involvement Is also Noted in “Non-Neuronopathic” GD1

GD1 has traditionally been defined as non-neuronopathic, and hence, no neurological symptoms or signs are evident. In 1980, Soffer et al. described autopsy findings of widespread perivascular clusters of Gaucher cells in cortical and subcortical regions in a 51-year-old man with GD1. While the affected blood vessels were surrounded by an intense fibrillary reaction, there was no neuronal loss and no accumulation of GlcCer in the brain. Importantly, despite the autopsy findings, the patient did not show any neurological symptoms [23]. In the case series by Wong et al., the primary neuropathological features described in the seven cases with GD1 were astrogliosis and perivascular Gaucher cells [22]. Affected brain regions in GD1 were described as gliotic with perivascular and fibrillary astrogliosis, evident by GFAP staining, but again, without prominent neuronal loss [22]. Hippocampal involvement was most prominent in the CA2 region, modest in CA3–4 and CA1 was typically spared.

Hulková et al. examined the frontal cortex and cerebellum in a 59-year-old woman with GD1 and reported occasional perivascular Gaucher cell clusters in white and grey cerebral matter and in the leptomeninges. Astrocytosis was noted in the white matter and subpial regions with mild gliosis in the dentate gyrus. In addition, lipofuscin particles were noted in Purkinje cells, Bergman astroglia, and cortical neurons, further documenting mild neuropathological involvement in GD1, without clinical neurological symptoms [24].

## 4. Neuropathologic Findings in Neuronopathic GD

Neuronopathic GD (nGD) encompasses GD2 and GD3, which both affect the central nervous system in several ways. Elevated levels of brain GlcCer and GlcSph occur in both types, although levels tend to be higher in GD2 [22,25,26,27]. Gaucher cells are also found in the brains of patients with both types, but there is some indication that their localization differs. In GD2, there can be free Gaucher cells in the cerebral cortex, with or without additional perivascular Gaucher cells. In GD3, however, perivascular Gaucher cells were predominant [28]. However, there is at least one case report of a patient with GD3 who was also found to have free parenchymal Gaucher cells [29]. Four neuronal alteration patterns have been suggested in patients with nGD: (1) mild and nonselective, (2) cerebellodentate, (3) bulbar, and (4) thalamocortical. Patterns (3) and (4) are common in patients with GD2, while pattern (1) is more characteristic of patients with the Norbottnian subtype of GD3 and pattern (2) is observed in other GD3 cases [30]. As described above, pyramidal neuronal loss in hippocampal layers CA2–4 is observed in nGD, with CA2 being the region most severely affected and CA1 largely spared [22]. Additionally, cortical laminar necrosis of the third and fifth cortical layers occurs in conjunction with astrogliotic neuronal loss of the fourth layer, though fourth layer abnormalities are largely localized to the occipital lobe [22,27].

Clinically, the initial distinction between GD2 and GD3 is the age at symptom onset. GD2 is diagnosed perinatally or in infancy while GD3 can present at any age, but often has a later diagnosis. GD2 has some unique presentations, including hydrops fetalis, congenital ichthyosis, severe stridor, and failure to achieve an independent gait [3]. In the case series by Wong et al., hippocampal involvement in GD2 was particularly severe, with significant neuronal loss. The few remaining hippocampal CA2 neurons observed were described as basophilic and shrunken [22]. The finding of particularly severe gliosis in CA2 in a GD2 case was also reported by Kaga et al. in an infant who died at six months. In this child, Gaucher cells were found both in the perivascular regions of the cerebrum and in the brainstem. Neuronal loss was observed in the brainstem, especially in nuclei of cranial nerves III, V, VII, and the superior olivary complex. The dentate nucleus, as well as the granular layer of the cerebellum, were lost [31].

An early study by Kaye et al. studied neuropathological differences between patients with GD2 and GD3. In the two cases of GD2 studied, GlcCer accumulation, Gaucher cells, gliosis, and microglial nodules were observed, and the level of GlcCer accumulation correlated with degree of neurodegeneration. The one case of GD3 reported displayed a similar pattern of GlcCer accumulation, but surprisingly lacked the other neuropathological findings [27]. Other studies have, however, found neuropathological changes in GD3, possibly reflecting the clinically diverse phenotypes collectively associated with GD3. In another case, a 10-month-old girl, clinically diagnosed with GD 3 and progressive stimulus-sensitive myoclonus, as well as bulbar signs, was studied. The patient showed widespread focal intraparenchymal Gaucher cells in cerebral cortex, mostly evident in lamina 4, as well as in the granular cell layer of the cerebellum. GFAP immunoreactivity indicating astrogliosis was increased in lamina 4 and, to a lesser extent, in lamina 2 in cortical samples, where mild to moderate neuronal loss was also evident. The pons, medulla oblongata, and substantia nigra (SN) all showed glial scars. In addition, severe loss of neurons and astrogliosis in the dentate nucleus and some loss of Purkinje cells were observed. Brain GlcCer levels were elevated both in the frontal cortex and cerebellum. While the clinical diagnosis was reported as GD3, the authors concluded that the neuropathological findings were a combination of the patterns expected in GD2 and GD3, highlighting the phenotypic spectrum in nGD [30].

Several studies of patients with GD3 have suggested that the dentate nucleus is the region most severely affected [29,32]. An autopsy report of a child with severe GD3 with a progressive generalized stimulus-sensitive and action myoclonus and cerebellar ataxia showed selective neurodegeneration of the cerebellar dentate nucleus and dentatorubrothalamic pathway [32]. The remaining neurons of the dentate nucleus showed signs of pyknosis and nuclear condensation. Loss of myelin and axonal profiles were also present in this neuronal population, along with a reduced number of fibers extending from the dentate nucleus. The fiber loss was selective to the superior cerebellar peduncle which includes the dentatorubrothalamic pathway [32]. Interestingly, neuronal populations in other brain regions, including the cerebral and cerebellar cortices, thalamus, basal ganglia, and inferior olivary nucleus, did not show evidence of decline or damage. There was only one focal ependymal lesion with infiltration of Gaucher cells observed, and specifically no loss of Purkinje cells. The authors concluded that the restricted dentate damage supports a central role of this nucleus in myoclonus. Furthermore, Alzheimer’s type 2 astrocytes were located in the basal ganglia, substantia nigra, and inferior olivary nucleus, implicating a systemic metabolic disorder [32]. In another early case report of a patient with stimulus-sensitive myoclonus, cerebellar ataxia, and general seizures, Winkelman and colleagues reported somewhat similar neuropathological findings, where the deep nuclei of the cerebellum were most severely affected. While there was no loss of Purkinje cells, mild astrogliosis was observed in the molecular layer of the cerebellar cortex. However, in this case signs of neuronophagia in the brainstem and multiple perivascular aggregates of Gaucher cells in the subcortical white matter were evident [33].

Burrow et al. performed a thorough neuropathological evaluation of a twelve-year-old child with GD3 who had been treated with enzyme replacement therapy for 11 years. Clinically, the child developed a cerebellar tremor, myoclonus, progressive ataxia, and generalized tonic–clonic seizures. At autopsy, isolated and nodular clusters of CD68 positive macrophages were seen throughout the cerebrum, often compressing arterioles. These perivascular clusters were also found in the basal ganglia, brainstem, hippocampus, cerebellum, and thalamus. Again, neuronal loss was prominent in the cerebellar dentate nucleus. A marked loss of Purkinje cells was noted. Diffuse astrogliosis was observed, often surrounding engorged macrophages. Phosphorylated Tau was identified in neuronal soma and processes in the hippocampus, basal ganglia, and cerebellum. In addition, rare cells in the cortex and hippocampus showed enhanced α-syn immunoreactivity. Thus, despite the patient’s young age, there were markers suggestive of a potential neurodegenerative disorder [29].

GD3 includes the “Norbottnian” subtype, named for a geographic isolate in northern Sweden where it was first described. This subtype, which is generally associated with *GBA1* genotype L444P/L444P, is characterized by infantile or juvenile onset, with slow progression of CNS involvement [34]. Conradi et al. conducted a morphological and biochemical analysis of five GD3 brains from this cohort, demonstrating Gaucher cells in each case. In two cases, loss of neurons and myelin near the Gaucher cells was observed. Varying degrees of neuronal loss, satellitosis (clustering of glia around neurons) [35], and neuronophagia were noted in all five patients. Light microscopy demonstrated lipofuscin with simple and complex lipids, but not glycolipids. Inclusion bodies were seen in both cerebral and cerebellar neurons, the dentate nucleus, and pons. GlcCer accumulation was present in these cases, although the levels varied. They tended to be higher in patients who had undergone splenectomy and were affected by the generalized lipid storage processes in specific individuals. Higher levels of GlcSph were noted in cases with more advanced nerve cell loss. This led to the suggestion that the accumulation of lipid substrates may act to prime a neurodegenerative process [22,34], which is one hypothesis proposed regarding why some patients with GD develop LBD [36].

## 5. Neuropathology of Gaucher-Associated Parkinsonism

The unanticipated link between the monogenetic disorder GD and the multifactorial neurodegenerative disorder PD has blurred the boundaries between genetic and sporadic Lewy body disorders (LBDs). Similar to sporadic LBD (sLBD), patients with GD who develop parkinsonism (GD-LBD) have a wide spectrum of phenotypes, ranging from slowly progressing L-DOPA responsive PD to rapidly progressive dementia with Lewy bodies (DLB) presentations [37]. While, on an individual basis, patients with LBD carrying *GBA1* mutations are clinically indistinguishable from those with sporadic disease, as a group, patients with parkinsonism who are either homozygous or heterozygous for *GBA1* mutations, have an earlier age of onset, faster progression, and more pronounced cognitive decline than those without mutations [37,38,39]. While the literature describing the neuropathology of heterozygous *GBA1*-carriers is expanding [40,41,42], there are only a few published neuropathological evaluations describing findings in homozygous patients with both disorders (summarized in Table 1). Unlike some of the other familial PD-related genes, at autopsy patients with *GBA1*-LBD regularly exhibit Lewy body (LB) pathology, mirroring a core neuropathological feature in PD and DLB [43]. LBs are neuronal perikaryal deposits mainly composed of misfolded α-syn. In addition, more than 80 different proteins, membranes, lipids, and distorted organelles have also been identified in these aggregates [44,45]. There are two subtypes of LBs, the classical brainstem type and the cortical type, each with a different localization, as well as a different microstructure, which affects the likelihood of their identification during the neuropathology examination [46]. Since *GBA1*-LBD shares essential histopathological features of LBD, the histopathological signature in GD-LBD could potentially provide insights into pathophysiology relevant to a larger group of affected patients.

In the early 2000s, Tayebi et al. published a case series describing patients with GD-LBD, suggesting that GCase deficiency may cause patients to be more vulnerable to parkinsonism. Brief neuropathological descriptions were included for four of the cases. Each exhibited a loss of dopaminergic neurons in the *substantia nigra pars compacta* (SN), the pathological hallmark for PD, as well as LB pathology, although the distribution of LBs varied among the patients. Specifically noted by the authors were brainstem-type LBs in the hippocampal regions CA2–4, sparing CA1 [10]. As noted above, these regions are specifically affected in GD. One year later, Wong et al. published additional neuropathological description of the same patients. Each of the GD-LBD cases exhibited astrogliosis in hippocampal areas CA2–4, the calcarine cortex layer 4b and the cerebral cortex layer 5, as reported in GD1 cases without parkinsonism. The SN showed neuronal loss, brainstem-type LBs, and gliosis. Two of the cases also had brainstem-type LB pathology in hippocampal pyramidal neurons, and in a third, brainstem-like LBs were limited to the SN. The fourth case had both brainstem and widespread cortical LBs consistent with diffuse LBD (Figure 2). The included cases had different *GBA1* genotypes, indicating that no specific mutation predisposes patients to LBD. Of the two homozygous N370S patients, only one had hippocampal LB involvement [22]. Even though neuropathology is the gold standard for the diagnosis of neurodegenerative disorders, there is, to date, no neuropathological criterion separating DLB from PD with dementia (PDD) [49,50]. DLB cases tend to have a larger LB burden, especially in the temporal lobe and CA2 region of the hippocampus, as well as elevated Alzheimer’s disease-related pathologies compared to PDD [51,52,53]. The degree of Lewy pathology in the hippocampal CA2 region has also been linked to cholinergic depletion and dementia development in PD [54]. It is tempting to correlate the involvement of Lewy pathology in hippocampal regions in GD-LBD with the more pronounced cognitive decline seen in *GBA1*-PD patients, but in asmuch as since hippocampal involvement is also observed in LBD without *GBA1*-mutations, more cases need to be examined to conclude whether the spread of LB pathology differs from sporadic cases [55].

Another source of neuropathological studies of GD-LBD now results from the inclusion of cases of GD in large autopsy series performed on subjects with PD. After genetic screening of a pathology cohort of more than 1200 patients with neurodegenerative disease, Blauwendraat et al. identified one case who was homozygous for *GBA1* N370S, as well as heterozygous for *LRRK2* G2019S. The patient presented clinically with PD and showed neuronal loss in the SN but exhibited no Tau or LB pathology [47]. Neuropathological reports of *LRRK2* G2019S carriers have heterogeneous results with regards to LB pathology, possible explaining the lack of LB pathology in this case [43]. Furthermore, Adler et al. examined 12 *GBA1*-carriers with PD looking to establish the neuropathological differences between *GBA1*-PD and sPD. This series included one case with genotype N370S/N370S, and hence GD-PD, but no individual information regarding neuropathological findings or comorbidities in this subject was reported [42].

It is still unclear whether there are histopathological differences between GD-LBD and sLBD. Several studies suggest a more widespread cortical LB burden in *GBA1*-PD, although this remains under debate [40,41,42,52,56]. While, as mentioned, there was one report of increased α-syn in both the hippocampus and cortex in a 12-year-old patient with GD3, no α-syn pathology was detected in five infants with GD2 who had widespread Gaucher cells in the CNS. This indicates that α-syn can accumulate early but suggests that GD disease burden does not directly correlate with α-syn pathology [29,57]. Since incidental LB pathology is found in approximately 10% of healthy people above 60 years old, α-syn accumulation in single cases should be interpreted with caution as no clinical features of parkinsonism were detected [58]. On a molecular level in a small cohort, Goker-Alpan et al. showed that GCase was present in LBs specifically in *GBA1*-LBD. In patients with GD-LBD over 80% of LBs stained positive for GCase, 33–90% in heterozygote carriers and <10% in sPD [59]. This suggests a role for mutant GCase in LB formation in *GBA1*-related disease, but as mentioned above, a large number of proteins have been found in LBs, many without a documented role in the pathophysiology. Furthermore, the variability in heterozygotes implicates additional individual stressors and/or protective factors. Unlike in nGD, in the few cases of GD-LBD investigated, as well as in heterozygous *GBA1*-carriers, no significant GluCer or GlcSph accumulation was observed in the CNS [10,60]. Slightly increased GluCer has been reported in the SN in sPD, although the significance of this finding is unclear [61].

The rare synucleinopathy multiple system atrophy (MSA) is marked by α-syn inclusions in oligodendrocytes as opposed to neurons. Whether there is a link between *GBA1* and MSA development is still not settled [48,62,63]. Interestingly, one case of autopsy-verified MSA was incidentally found to be homozygous for N370S but was never diagnosed with GD during life. Examination showed atrophy with neuronal loss and gliosis of the basal ganglia and cerebellum, indicating a mix between the major subtypes of MSA, MSA-P (parkinsonian), and MSA-C (cerebellar). The patient had glial cytoplasmic inclusions with α-syn, indicative of MSA, and rare neurofibrillary tangles, seen in Alzheimer’s disease, but no LB pathology was evident [48].

## 6. Discussion and Conclusions

The appreciation of the link between *GBA1* and the LBDs has stimulated an upsurge in *GBA1*-related research activity. However, the field is hampered by the dearth of well-documented autopsy studies, and the full spectrum of neuropathological findings associated with GD has yet to be established. Many of the reported cases exhibit astrogliosis in cortical layers 3 and 5 and in hippocampal regions CA2–4. Selective neuronal loss is described in neuronopathic GD. Rare autopsy studies of GD3 show limited depletion of neurons in the dentate nucleus. Findings regarding loss of Purkinje cells are conflicting, as is the degree of Gaucher cell infiltration. The variability in pathological findings likely reflects the recognized clinical heterogeneity in GD. However, it should be noted that these pathological patterns are based on very few cases and certainly do not cover the entire phenotypic spectrum of GD. Furthermore, the specific brain regions examined, and the staining techniques used vary among publications, limiting direct comparisons.

Brain accumulation of GlcCer and GlcSph in nGD has been observed. While generally seen as toxic and causative of neurodegeneration, this does not provide an explanation regarding the selective neurodegeneration which occurs as a response to the systemic increase. In control brain, Wong et al. demonstrated increased levels of GCase localized to brain regions specifically affected in nGD [22]. Studies investigating the causes of the cell type specific vulnerability could increase our understanding of the role of the implicated lipids in LBD and other neurodegenerative diseases, including Alzheimer’s disease.

Several of the original neuropathological studies in GD patients were published before the link to LBD was known, and therefore, many of the cases were not specifically examined for α-syn pathology. Interestingly, the GD-affected hippocampal areas CA2–4 have also been specifically involved in GD-LBD. Hippocampal involvement is common in sPD according to Braak staging [55]. Recently, the question of whether Braak staging is applicable to all subtypes of PD has been raised, and hence, more studies of various well-defined PD cohorts are needed [64,65]. While neuropathological assessments of GD-LBD are quite limited, the similar LB pathology reported suggests that GD-LBD might provide a relevant model to understand cellular pathways generally relevant to LBD. Careful *GBA1* genotyping of brain bank PD series could potentially lead to the identification of further cases. However, without a better understanding of GD-related pathology, potentially important subtle differences between GD-LBD and sPD that could provide critical mechanistic clues might be overlooked.

## Figures and Tables

**Figure 1 ijms-23-05842-f001:**
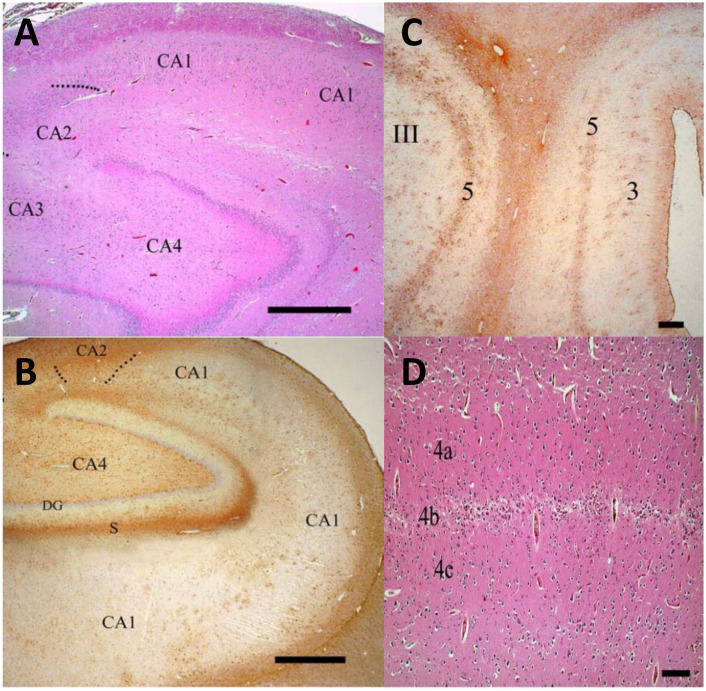
Regions affected in Gaucher disease (GD) include hippocampal regions CA2–CA4 and cortical layers. (**A**) Neuronal loss and astrogliosis observed in an autopsy of infant with GD2 case demonstrating astrogliosis in hippocampal regions CA2–CA4 with relative sparing of CA1 (H&E, 40× magnification). (**B**) Mild astrogliosis in GD1 detected by GFAP staining, without neuronal loss. (GFAP immunoperoxidase, 40× magnification). (**C**) Astrogliosis in GD1 shown by GFAF staining in cerebral cortical layer 5 (GFAP immunoperoxidase, 40× magnification). (**D**) Calcarine cortex in an infant with GD2 demonstrating neuronal loss and astrogliosis in layer 4b, sparing layer 4a and 4c (H&E, 40× magnification). Scale bars: (**A**) 1 mm; (**B**) 1 mm: (**C**) 250 μm; (**D**) 100 μm. Adopted with permission from Wong et al., 2004, Elsevier Inc. [22].

**Figure 2 ijms-23-05842-f002:**
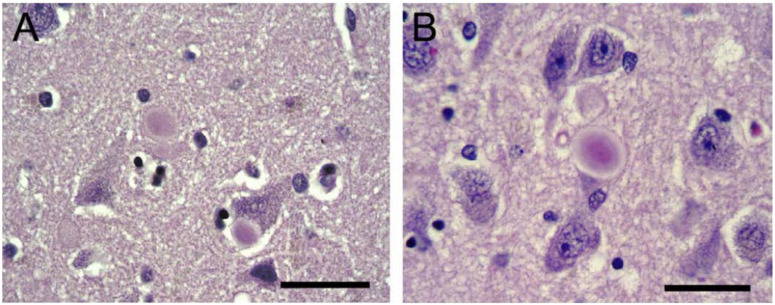
Intraneuronal inclusions similar to brainstem-type Lewy bodies identified in hippocampal neurons in regions CA2 (**A**) and CA3 (**B**) in a patient with both GD and parkinsonism H&E, 400X magnification. Scale bars (**A**) 35 μm; (**B**) 38 μm. Reprinted with permission from Wong et al., 2004, Elsevier Inc. [22].

**Table 1 ijms-23-05842-t001:** Autopsy studies of patients with both Gaucher disease (GD) and parkinsonism.

Report	Gender/ Age of Death	Gaucher Diagnosis	Genotype	Neuronal Loss in SN	LB Pathology	Clinical Features
Wong et al., 2004 [22]	F/62	GD1	N370S/unknown	Yes	Diffuse cortical LB	Parkinsonism and dementia
Wong et al., 2004 [22]	F/53	GD1	D409H/L444P+duplication	Yes	Brainstem LB in hippocampal CA2–4	Parkinsonism and dementia, supranuclear gaze palsy
Wong et al., 2004 [22]	M/75	GD1	N370S/N370S	Yes	Brainstem LB in SN	Parkinsonism and dementia
Wong et al., 2004 [22]	M/54	GD1	N370S/N370S	Yes	Brainstem LB in hippocampal CA2–4	Parkinsonism and dementia
Adler et al., 2017 [42]	F/73	Unknown	N370S/N370S	Unknown	Unknown	Parkinsonism and dementia
Blauwendraat et al., 2019 [47]	F/~90	Un-diagnosed	N370S/N370S	Yes	No	Parkinsonism
Sklerov et al., 2017 [48]	?/63	Un-diagnosed	N370S/N370S	Yes	No	Orthostasis, parkinsonism, cognitive deficits

## Data Availability

Not applicable.

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
