# Peer review of "Neuropathological Features of Gaucher Disease and Gaucher Disease with Parkinsonism"

_ijms, 2022, doi:10.3390/ijms23105842_

Round 1

Reviewer 1 Report

The work "Neuropathological Features of Gaucher Disease and Gaucher Disease with Parkinsonism" by Furderer et al. is an extremely interesting report, which, however, requires some significant changes:
- please explain the "+" sign next to the name of Ellen Sidransky1 + in the 4 lines of the text;
- on line 7 there is the phrase "* These authors contributed equally to this work", however, on line 8 the corresponding author is marked with the same * sign. Please change the tagging as it is confusing.
- all gene names quoted in the paper should be in italics, eg line 30, the GBA1 gene;
- all citations in the publication should be cited in brackets [...] and not (...) in accordance with the requirements in the journal format. Please, correct this throughout the text.

- I also ask you to read again the text and replace the quoted authors in the form of the name and year of the authors with the quotation in the form of a number that corresponds to it in the bibliography. Such inaccuracies appear on lines 55, lines 109-110, lines 153, or lines 288-289.
- Is it possible to provide a specific scale used in the photos in the presented figures (figures 1 and 2)?
- In table 1, please provide numerical citation.

Author Response

Comments to reviewers IJMS-1732348

Reviewer 1:

  1. Please explain the "+" sign next to the name of Ellen Sidransky1 + in the 4 lines of the text;
    on line 7 there is the phrase "* These authors contributed equally to this work", however, on line 8 the corresponding author is marked with the same * sign. Please change the tagging as it is confusing.

Thank you for noticing the mistake in the author list. We have clarified the symbols to avoid confusion.

  1. All gene names quoted in the paper should be in italics, eg line 30, the GBA1 gene.

We have edited the text accordingly.

  1. All citations in the publication should be cited in brackets [...] and not (...) in accordance with the requirements in the journal format. I also ask you to read again the text and replace the quoted authors in the form of the name and year of the authors with the quotation in the form of a number that corresponds to it in the bibliography. Such inaccuracies appear on lines 55, lines 109-110, lines 153, or lines 288-289.

The manuscript has been edited accordingly, however mostly without “track changes” due to the large number of changed characters. In the figure legends for Fig. 1 and Fig. 2 we kept the full citation as the pictures are directly reprinted or only slightly modified from the original publication. However, we added the bibliography number for clarity. Thank you for the thorough comment.

  1. Is it possible to provide a specific scale used in the photos in the presented figures (figures 1 and 2)?

Unfortunately, the µ symbol got misplaced in the editing phase. The correct scale bars can now be found in the figure legends.

  1. In table 1, please provide numerical citation

Table 1 has been redrawn and now includes the missing information. 

Reviewer 2 Report

The manuscript by Furderer and colleagues describes a comprehensive overview of the neuropathological aspects in Gaucher disease(GD) and GD-associated neurodegenerative forms (LBD and PD). The manuscript is well written and provides a critical discussion of the literature related to the topic. There are only few minor adjustments required to improve the quality of the manuscript:

1) Line 38-39. This sentence has a problem and needs to be rephrased.

2) Line 53. I would change "developed" into "develop" as the parkinsonian features are still exhibited by currently affected patients.

3) Table 1 needs to be redrawn. References numbers and year of publication are missing . Addditionally, I would recommend to set the current columns size as the text is wrongly justified for each term.

4) Line 338 and line 342: there are wrong commas (for instance after Purkinje cells)

5) Line 348: the reference for Wong et al., is missing

6) Line 353 the "alpha" is missing for alpha-syn.

Author Response

  1. Line 38-39. This sentence has a problem and needs to be rephrased and in Line 53: I would change "developed" into "develop" as the parkinsonian features are still exhibited by currently affected patients

Thank you for your comment, the two sentences have been rephrased for clarification.

  1. Table 1 needs to be redrawn. References numbers and year of publication are missing. Additionally, I would recommend to set the current columns size as the text is wrongly justified for each term.

Table 1 has been redrawn to include the missing information.

  1. Line 338 and line 342: there are wrong commas (for instance after Purkinje cells), line 348: the reference for Wong et al., is missing and Line 353 the "alpha" is missing for alpha-syn.

The mistakes above have been changed. We are grateful for the careful evaluation of the manuscript.